# Ragweed Major Allergen Amb a 11 Recombinant Production and Clinical Implications

**DOI:** 10.3390/biom13010182

**Published:** 2023-01-16

**Authors:** Tudor-Paul Tamaș, Maria-Roxana Buzan, Lauriana-Eunice Zbîrcea, Monica-Daniela Cotarcă, Manuela Grijincu, Virgil Păunescu, Carmen Panaitescu, Kuan-Wei Chen

**Affiliations:** 1Center of Immuno-Physiology and Biotechnologies, Department of Functional Sciences, Victor Babes University of Medicine and Pharmacy, 300041 Timișoara, Romania; 2OncoGen Center, “Pius Brînzeu” Timiș County Clinical Emergency Hospital, 300723 Timișoara, Romania

**Keywords:** recombinant Amb a 11, ELISA, RBL assay, clinical features

## Abstract

Ragweed pollen is highly allergenic and elicits type I hypersensitivity reactions in the exposed populations. Amb a 11 is a recently discovered component of this pollen, and its biological role in allergy is still being researched. In our study, ragweed allergy patients were recruited prospectively over a three-year period; a comprehensive questionnaire was administered, and sera were collected and stored. The production of recombinant Amb a 11 was achieved in parallel with patients’ recruitment. The gene coding for mature protein was inserted in *E. coli* and in Sf9 *Spodoptera frugiperda* cells. The recombinant allergens (designated eAmb a 11 and iAmb a 11) were tested for His-tag presence in Western blot. IgE reactivity was evaluated in 150 patients’ sera for both recombinant allergen forms in ELISA, with 5 positive sera being tested further by hRBL (humanized rat basophilic leukemia) hexosaminidase release assay. Both allergen forms were proven to be IgE-reactive His-tagged proteins, with an extensive overlap of positive sera (92 toward the former recombinant allergen, 100 toward the latter) and an overall Amb a 11 sensitization prevalence estimated at 68.67%. The hRBL mediator release assay revealed a significant, slightly weaker effect of recombinant allergens when compared with nAmb a 1. Sensitization to this major allergen appears to be associated with more severe asthma symptoms (OR = 4.71, 95% CI = 1.81–12.21). In conclusion, recombinant Amb a 11 is a bona fide allergen, which is IgE-reactive and an inducer of hRBL degranulation. It is an important IgE-reactive component from ragweed pollen, with high IgE sensitization prevalence in the sample population and allergenicity of the recombinant allergen comparable to Amb a 1.

## 1. Introduction

Common ragweed (*Ambrosia artemisiifolia*) is an annual herbaceous flowering plant native to North America and successfully acclimated, since the 19th century, in every other habitable continent [1,2,3]. Ragweed pollen is a common cause of seasonal allergy in North America, while in Europe it gains a prominent role for being reported as the outdoor allergen with the highest air concentration in Romania [4]. It is estimated that by 2050, the number of European ragweed allergy patients may double as the plant conquers new habitats [5,6]. Ragweed allergy generates increasing medical costs and reduced quality of life (QoL) [7]. The large spread of this allergenic pollen and the wide range of associated health issues constitute a public health concern, recognized in patient education, ragweed awareness, and eradication campaigns promoted in many affected areas from Europe to North America [2].

There is a wide variety of IgE responses in ragweed-pollen-allergic patients, with 12 allergens enrolled in the WHO IUIS database and numerous candidate allergen molecules discovered [8,9,10,11]. A recently discovered major allergen is Amb a 11, with dual biological function: protease and inducer of type I hypersensitivity responses. Other related proteins were described as major allergens in fruits and mites [8,9,12,13].

Natural Amb a 11 is a cysteine protease assigned to the C1A subfamily of proteases. This family includes proteases from plants, which are synthesized as inactive or lightly active proenzymes (pro-forms). The mature form was postulated to derive from the pro-form by an autocatalytic mechanism of maturation (proteolytic cleavage). Mature Amb a 11 is a single-chain protein with a calculated MW at 28 kDa, consisting of 262 amino acids with a calculated isoelectric point of 5.72. The optimal pH for protease activity was determined to be at 8–9 pH units (and over 50% relative activity at pH between 7 and 9.5). Structural characterization revealed a catalytic triad (Cys-155, His-289, and Asn-310) which is shared with papain and other members of its family. Natural Amb a 11 analysis by mass spectrometry revealed at least 20 isoforms and glycoforms, but it was established that glycosylation is not a requirement for IgE binding [14,15].

Our research aim was to evaluate the in vitro allergenic potential of recombinant Amb a 11 in terms of antigen recognition with patients’ sera, as well as basophil activation, which is a close model for the in vivo allergic response. This facilitates a more accurate assessment of the allergen’s clinical significance in patients from Western Romania—a hot spot for ragweed invasion in Southeast Europe. 

## 2. Materials and Methods

### 2.1. Study Group and Patient Recruitment

Adult ragweed-allergic patients (age ≥ 18 years) from Western Romania, Timișoara Metropolitan Area (TMA), referred to Allergologists between 2017 and 2019, were recruited. Patients were included based on known or newly discovered ragweed pollen allergy, by clinical diagnosis confirmed with positive common ragweed skin prick test (SPT) and/or sIgE. Medical procedures (skin prick test, peripheral blood collection) and experimental protocols with patient sera were performed after obtaining written informed consent. Ethical approval was obtained prior to study initiation from the Ethics in Scientific Research Commission of the “Pius Brînzeu” Timiș County Clinical Emergency Hospital, Timișoara (approval nr. 102/10-January-2017), which complies with the Romanian law, the EU Directive 2005/28/EC, and the Helsinki Declaration. 

An 18-item interviewer-administered questionnaire, comprising closed and open-ended questions, related to patient demographics, exposure to environmental factors, clinical information, and context, as well as self-assessment of disease impact, was completed for the selected patients. 

The peripheral blood samples were centrifuged, and sera were stored at −80 °C. 

### 2.2. Laboratory Supplies

Buffer components, protein electrophoresis and Western blot materials and reagents, and dyes were purchased from international suppliers (Merck, Darmstadt, Germany; Sigma-Aldrich, St. Louis, MO, USA; Gibco, Thermo Fisher Scientific, Waltham, MA, USA; Roche Diagnostics GmbH, Mannheim, Germany). nAmb a 1.01 was kindly provided by Biomay AG, Vienna Competence Center, Austria. rDer p 2 was kindly provided by the Department of Pathophysiology and Allergy Research at the Medical University of Vienna, Austria.

### 2.3. Recombinant Allergen Expression, Purification, and Detection

For the production of recombinant Amb a 11, we selected two expression systems—*E. coli* and *Sf9* insect cells. The entire Amb a 11 sequence (pre-pro Amb a 11, consisting of 368 amino acid residues, GenBank entry AHA56102.1) was initially selected for expression [16,17]. Because of a low production yield and IgE reactivity of the proenzyme (14% on preliminary ELISA testing), we decided to express the mature cysteine protease (262 amino acid residues) in both *E. coli* and *Sf9* insect cell expression systems. 

A codon-optimized construct with a C-terminal hexahistidine tag was designed for protein expression in *Sf9* insect cells (Gibco, Thermo Fisher Scientific) using the pTM1 vector via BamHI/SmaI site and E. coli BL21-Gold(DE3) cells (Agilent Technologies, Santa Clara, CA, USA) using pET27b via the NdeI/XhoII site (ATG: biosynthetics, Merzhausen, Germany). 

For *E. coli* expression, after heat shock transformation, the cells were plated on LB agar w/50 µg/ml kanamycin and then cultured overnight in LB medium. Large-scale expression in 1 L of LB was induced by adding 0.5 mM isopropyl-thiogalactopyranoside IPTG (Carl Roth GmbH, Karlsruhe, Germany) once the culture reached an OD600 ≈ 0.3 measured on a microplate reader (Tecan infinite M200 Pro, Grödig, Austria). After overnight incubation with IPTG (120 rpm, 35 °C), the culture was centrifuged, and the cell pellet was re-suspended in 15 mL of chemical lysis buffer: 25 mM imidazole, 0.1% Triton-X 100, pH = 7.4. The cells were disrupted by freeze/thawing three times, in liquid N_2_ followed by 50 °C water bath, and centrifuged after the addition of 1µL DNase (10 U/µL, Roche Diagnostics GmbH). The resulting pellet was solubilized in urea buffer B (8 M urea, 100 mM NaH_2_PO_4_, 10 mM Tris, pH = 8) overnight at RT on a magnetic stirrer, then centrifuged, and the protein-rich supernatant was purified using NINTA: Ni^2+^ NTA affinity chromatography (QIAGEN, Hilden, Germany). The significant elutions were pooled and dialyzed sequentially (for urea removal) toward 10 mM NaH_2_PO_4_, pH = 7.4.

For insect cell expression, *E. coli* DH10 competent cell (Thermo Fisher Scientific) transformation with plasmid DNA was performed by the Bac-to-Bac Baculovirus Expression System protocol from Invitrogen (Thermo Fisher Scientific). The bacteria were then plated on LB agar /10 µg/mL tetracyclin, 50 µg/ml kanamycin, 7 µg/mL gentamycin, 40 µg/mL IPTG, and 100 µg/mL X-Gal. White colonies were re-plated and screened for bacmid insertion by PCR: cells from each colony were resuspended in 10 µL ultrapure water, mixed with 25 µL GoTaq G2 Hot Start Green Mastermix (Promega, Madison, WI, USA) and 2.5 µL of each primer: M13 Forward (5′-CCCAGTCACGACGTTGTAAAACG-3′) and Reverse (5′-AGCGGATAACAATTTCACACAGG-3′). A colony containing the insert was then cultured overnight (LB medium, 37 °C). The bacmid was isolated the next day with a Midiprep kit (Promega, Madison, WI, USA). The DNA concentration was measured by NanoDrop 1000 (Thermo Fischer Scientific), and the aliquots were stored at −20 °C. 

*Sf9* cells were cultivated in Sf-900 SFM II (FBS-free medium) supplemented with 2.5% FBS, 10 µg/mL gentamicin, and 250 ng/mL amphotericin B. The cells were transfected using 2 µg of purified bacmid DNA for 2 × 10^6^ cells/2 mL medium/well in 6-well plates (Corning, New York, NY, USA). First, the cells were added to the plate for adherence (30 min, 27 °C). The FBS-containing medium was replaced with 2 mL SFM w/antibiotics, plus 100 µL transfection mix: 2 µg bacmid DNA, 6 µL FuGENE HD transfection reagent (Promega, Madison WI, USA), up to 100 µL nuclease-free water. The plates were incubated for 5 h at 27 °C. Afterward, the transfection medium was replaced with antibiotics and FBS-supplemented SFM and cultivated at 27 °C. After 72 h, the P1 baculovirus stock was harvested by the centrifugation of the pooled wells’ content and used for generating P2 and P3 baculovirus stock in successive 72 h incubations. The optimal expression conditions were determined in a small-scale experiment: 50 µL of baculovirus stock P3/1 × 10^6^ cells in SFM, 72 h culture. After centrifugation, the supernatant was dialyzed overnight against buffer A (50 mM NaH_2_PO_4_, 300 mM NaCl, 10 mM imidazole, pH = 8) and purified by NINTA (under native conditions according to the manufacturer; QIAGEN, Appendix A). The elutions with the highest amount of protein on 14% SDS-PAGE were pooled and dialyzed sequentially in order to remove the imidazole toward 10 mM NaH_2_PO_4_, pH = 7.4. The size and purity of the recombinant protein in the final buffer were verified on SDS-PAGE under reducing (R) (w/ß-mercaptoethanol) and non-reducing conditions (NR) (w/o ß-mercaptoethanol), with PageRuler Plus Prestained Protein Ladder (Thermo Fisher Scientific). Due to the low concentration of the recombinant allergen (iAmb a 11) in the supernatant from further larger-scale cultures, determined with Thermo Scientific™ Pierce BCA Protein Assay Kit, a decision was made to retrieve the protein by cell lysis, resulting in up to a 5-fold increase of protein yield. Briefly, this was achieved by the chemical lysis of the *Sf9* cells’ pellet, followed by three freeze–thaw cycles, then centrifugation. The sediment was placed in urea buffer B on a magnetic stirrer overnight at RT. Following centrifugation, the protein was separated from the supernatant by NINTA (under denaturing conditions, QIAGEN, Appendix A). The significant elutions were pooled and dialyzed sequentially, toward 10 mM NaH_2_PO_4_, pH = 7.4. The size and purity of the recombinant protein in the final buffer were verified on SDS-PAGE. 

### 2.4. Western Blot Protocol: His-Tag Detection and IgE Binding Detection with Patient Sera

After running the recombinant proteins on SDS-PAGE and blotting [18,19], protein bands were revealed with Ponceau S. Strips of PVDF membrane, 4 mm wide and 8 cm long, were cut and were washed and blocked with Gold buffer (40 mM Na_2_HPO_4_, 0.6 mM NaH_2_PO_4_, pH = 7.5, 0.5% Tween 20, 0.5% BSA, 0.05% NaN_3_). 

His-tag detection was performed by mouse anti-His epitope antibody (Dianova, Hamburg, Germany) diluted 1:1000 in Gold buffer (GB). After overnight (ON) incubation at 4 °C on shaker, alkaline phosphatase (AKP) labeled rat anti-mouse IgG1 (clone X56, BD Biosciences Pharmingen, San Jose, CA, USA), diluted 1:1000 in GB, was added. After washing three times with GB and two times with AP buffer (100 mM Tris, 100 mM NaCl, 10 mM MgCl_2_, pH = 9.5), blots were detected with 5-bromo-4-chloro-3-indolyl phosphate(BCIP)/Nitro blue-tetrazolium(NBT) in 1 mL AP buffer/strip.

For IgE binding detection, patient sera were added on each strip, diluted 1:10 in GB (1 mL/strip), and incubated overnight (ON) at 4 °C on shaker. The next day, after washing three times with GB, AKP labeled mouse anti-human IgE (clone G7-26, BD Biosciences, Pharmingen, San Jose, CA, USA), diluted 1:1000 in GB, was added, 1 mL/strip, then incubated ON at 4 °C on shaker. The following day, blots were detected using the same detection protocol as above for His-tag detection.

### 2.5. IgE Binding Frequency Measurement for Purified Recombinant Allergens

The IgE reactivity of the recombinant allergens was tested in ELISA against 150 ragweed-allergic patient sera. Recombinant allergens, or rDer p 2-positive control (5 µg/mL), were incubated overnight at 4 °C on 96-well plates (Maxisorp Nunc, Thermo Fisher Scientific). Plates were washed twice with PBST (136 mM NaCl, 2.6 mM KCl, 10 mM Na_2_HPO_4_, 1.7 mM KH_2_PO_4_, pH = 7.4, 0.05% Tween) and then blocked with PBST/3% BSA for 3 h at RT. Patient serum was diluted 1:5 in PBST/0.5% BSA, with 100 µL added in duplicate to the allergen-coated plates, then incubated at 4 °C overnight. After washing five times with PBST, horseradish peroxidase (HRP) labeled polyclonal goat anti-human IgE antibody (SeraCare, Milford, MA, USA), diluted 1:2500 in PBST/0.5% BSA, 100µL/well, was added, with plates incubated at 37 °C and 4 °C (45’ each). Five more washes with PBST followed, then the detection substrate—2,2′-Azino-bis(3-ethylbenzothiazoline-6-sulfonic acid) diammonium salt (ABTS) in 60 mM citric acid, 77 mM Na_2_HPO_4·_2H_2_O, and 3 mM H_2_O_2_—was added. The absorbance was measured at a 405 nm vs. 490 nm reference wavelength on a microplate reader (Tecan infinite M200 Pro, Grödig, Austria). Negative controls were used for the calculation of statistical cut-off (mean + 3SD) to identify the positive sera. Furthermore, a set of 30 ELISA-positive sera were investigated for CCD reactivity. A possible difference in IgE reactivity between iAmb a 11 from culture supernatant and the one retrieved by cell lysis was also explored by ELISA, with the latter being used henceforth.

### 2.6. Humanized Rat Basophil Leukemia (hRBL) Cell Cultivation and Mediator Assay

The allergenic activity of recombinant allergens was evaluated by hRBL (clone RS-ATL8) degranulation assay. The hRBL cells were kindly provided by Dr. Ryosuke Nakamura (National Institute of Health Sciences, Tokyo, Japan) [20] and were cultivated in a cell culture medium MEM supplemented with 10% heat-inactivated FBS, 100 U/mL penicillin–streptomycin, 0.2 mg/mL geneticin, 0.2mg/mL hygromycin B, 0.2 mM L-Glutamine. For the assay, hRBL cells were loaded with heat-inactivated patient serum diluted 1:10 in a cell culture medium on 96-well plates incubated overnight at 37 °C. Eight recombinant allergen concentrations (10 µg/mL–1 pg/mL), with nAmb a 1.01 as a reference, in triplicates, were used the next day for stimulation. Complete ß-hexosaminidase release was induced by the addition of 10% Triton-X on hRBL cultured with patient serum, without adding allergens, on each plate. The ß-hexosaminidase release was detected by adding 4-methyl-umbelliferyl-N-acetyl-β-D-glucosaminide and measuring fluorescence at 465 nm (excitation wavelength = 360 nm) on a Thermo Scientific™ Varioskan™ LUX Multimode Microplate Reader.

### 2.7. Streptavidin ImmunoCAP

For custom ImmunoCAP, recombinant allergen was dialyzed against 1 L of 0.1 M NaHCO_3_, 1 M NaCl buffer. After dialysis, the protein concentration was measured. Biotin stock 10 mg/mL was used in 5-fold molar excess to bind the recombinant allergen (3 h, RT, mixed), followed by PBS dialysis and another measurement of the protein concentration. The biotinylated allergen was bound on Streptavidin ImmunoCAP o212 (Thermo Fisher Scientific) using a custom program (prewashed, loaded 50 µL of 100 µg/mL allergen/CAP, incubated 30’, washed) on a Thermo Scientific™ Phadia100, then detected with patient sera in a Thermo Scientific™ Phadia250 instrument according to manufacturer protocols.

### 2.8. Data Collection and Processing

Data were recorded in Microsoft Excel using Microsoft 365 suite (first access date: 7 February 2021). Statistical processing was performed and graphs were drawn using online tools, Microsoft Excel, MedCalc version 20.113, GraphPad Prism version 9.1.0 (221), imageJ version 1.53k. Statistical methods included: *t*-test for parametric comparison, exact Wilcoxon–Mann–Whitney test for scores, Chi-squared test for frequencies, Z-test for percentages, regression analysis [21,22].

## 3. Results

### 3.1. Protein Features on SDS-PAGE and His-Tag Detection

Both expression systems resulted in His-tagged proteins with similar MW on SDS-PAGE, with a main band just below 36 kDa and a secondary band at 72 kDa (Figure 1A). The eAmb a 11 protein appears to be slightly below iAmb a 11 on gel, at 36 kDa. On the 14% polyacrylamide gel, faint bands are visible just below 72 kDa (above the main band) and also very faintly below 28 kDa, 17 kDa, and above 10 kDa. SDS-PAGE, together with the gel filtration evidence of two protein fractions (Appendix A, unpublished material), suggests that Amb a 11 in both expression forms can aggregate preferentially as dimers, which display the hexahistidine tag on the surface. The two recombinant proteins are positive for His-tag, as shown in blot detection with anti-His-tag antibodies (Figure 1B), with stronger binding under non-reducing conditions for the proteins and their dimers. Weak bands also appear in the lower part, mainly in the eAmb a 11 blot, thereby suggesting the presence of His-tagged proteins below 28 kDa and below 17 kDa.

### 3.2. Western Blot against Patient Sera 

The IgE reactivity of the recombinant proteins was tested on Western blot (Figure 2). Thus, both rAmb a 11 forms and their dimers exhibit IgE reactivity when tested against the patients’ sera. The small fragments demonstrated in SDS-PAGE and His-tag detection do not appear to be IgE-reactive, while the main protein band at 36 kDa is prominent, along with a weak band just below 72 kDa.

### 3.3. ELISA 

The ratio of Amb a 11-positive sera determined by ELISA (Figure 3) was 61.33% for eAmb a 11 and 66.67% for iAmb a 11, with a 68.67% overall calculated serum prevalence of IgE sensitivity toward the recombinant allergens. There was a consistent trend toward lower ELISA signal elicited by iAmb a 11 vs. eAmb a 11 in all the plates, when compared pairwise (same sera) between wells coated with each recombinant allergen on the same plate. Such a trend was confirmed by paired *t*-test results (*p* < 0.05) in five of the eight ELISA plates. The threshold for positive results (calculated from negative control mean + 3SD) was also lower in the case of iAmb a 11 results.

### 3.4. RBL Degranulation Assay

The recombinant proteins’ relevance as in vivo allergens was closely approximated by hRBL assays, with sera that previously tested positive in ELISA for IgE reactivity against these allergens, in five cases. The response toward Amb a 11 recombinant forms was compared with the response toward nAmb a 1. Different response features (quantified as RBL degranulation percentage) can be noted, from a maximum Amb a 11-induced degranulation of 77% in patient C (a stronger response even than nAmb a 1-induced degranulation, at low allergen concentrations) to a medium intensity response in patients A and B (54% and 45% maximum degranulation) and a low-intensity response in patients D and E (with 15% and 27% maximum degranulation) (Figure 4).

### 3.5. ImmunoCAP Testing

Streptavidin ImmunoCAP was assayed using the recombinant protein iAmb a 11 coupled with biotin. The test showed only very few (3) positive results recorded on the pilot ImmunoCAP run with 44 ELISA-positive sera toward iAmb a 11 (Table 1). A similar result was obtained for the eAmb a 11 recombinant protein (not published).

### 3.6. Statistical Analysis

Among the 150 patients, there is a high prevalence of allergic disease symptoms, virtually all present with rhinitis, and a significant percentage is associated with conjunctivitis, asthma, and/or urticaria (Table 2). A family history of allergy is present in almost a third of the cases, while cigarette smoke represents the most frequent environmental exposure. Most of the patients are polysensitized, with HDM as a second allergen source. Asthma-related covariates were identified from the questionnaire as follows: age, sex, family history and allergy history (years), environmental exposures (domestic or professional), and other allergies, and entered into a regression model to evaluate their association with asthma-like symptoms (wheezing, cough, chest tightness, or shortness of breath) in the study group.

Several statistically significant differences were noted when comparing Amb a 11-positive and -negative patient data in regard to gender distribution, exposure to pets, polysensitized status, SPT reactivity pattern, and asthma symptom scores (Table 2 and Table 3). In the logistic regression model, IgE sensitization to Amb a 11, positive dog SPT, and domestic exposure to cats were identified as risk factors for more severe asthma or asthma-like symptoms (Table 3). The QoL indicators were slightly more impaired in the Amb a 11-positive group, but the differences did not reach statistical significance (Table 2).

## 4. Discussion

Recent research on recombinant Amb a 11 *E. coli* expression resulted in the structural characterization and evidence of mature Amb a 11 in SDS-PAGE. A distinct band was reported at 35 kDa for the mature protein obtained by the refolding of the *E. coli*-expressed proenzyme (hence, non-glycosylated). It was also established that glycosylation is not a condition for IgE binding [15]. 

In another proteomic study, 2D-PAGE spots containing Amb a 11 were detected below 40 kDa using pooled sera from ragweed-allergic patients [23]. It was reported as a purified natural Amb a 11 protein, migrating to 37 kDa on the gel. Thus, there appears a difference between the glycosylated (natural) protein and the non-glycosylated form previously described. This lends support to the hypothesis that our iAmb a 11 recombinant allergen is glycosylated, likely due to the presence of an N-linked glycan [24].

The very weak bands of low molecular weight appearing on SDS-PAGE are likely the products of recombinant allergens’ limited proteolysis, in an autocatalytic manner. The lack of His-tag fragments in the area above 10 kDa suggests that those are N-terminal peptides, while the C-terminal cleavage products below 28 and 17 kDa appear to hold the hexahistidine tag. Yet, these cleavage products, both C-terminal and N-terminal, are not IgE-reactive, indicating that their secondary structure was disrupted as a result of proteolysis. In this case, glycosylation could decrease the rate of proteolysis for iAmb a 11 in a manner similar to the one reported for other allergens [25]. This could explain the relative scarcity of low molecular weight bands in the iAmb a 11 gel and blot compared to eAmb a 11. As the recombinant proteins are stored at −20 °C, from longer-term observations, they appear to be fairly stable.

Amb a 11 in aqueous solution (10 mM NaH_2_PO_4_, pH = 7.4) appears to spontaneously form Der p 1-like non-covalent dimers [26]. The dimers are positive for His-tag and are also IgE-reactive. As dimerization is a proven enhancer for the allergen’s ability to cross-link membrane-bound IgE, with stronger in vivo effects, it highlights the strong biological activity of this ragweed allergen [27]. 

Previously, the reported prevalence of IgE sensitization to Amb a 11 in ragweed pollen allergy was 63.5% (14 of 22 patients) using 2D gel electrophoresis (PAGE) mapping in Hungarian patients or 54% (50 of 92) in a mixed group of European/American patients using the same technique. IgE reactivity against purified nAmb a 11, in non-denaturing dot-blot conditions, was confirmed for 66% of patients [14,23]. Our results reveal Amb a 11 as a major allergenic component from ragweed pollen, with 68.67% IgE binding prevalence in the study group, as determined by ELISA.

In ELISA, the wells were coated with the same amounts of both recombinant allergens. The trend toward lower absorbance in iAmb a 11 wells may be the effect of steric hindrance by an oligosaccharide side-chain in iAmb a 11, which resulted in less effective antibody binding [28,29]. Nevertheless, there is a slightly higher prevalence of IgE reactivity against iAmb a 11, as determined by ELISA, which could be explained by glycosylation and better folding in the case of insect-cell-expressed vs. E. coli-expressed proteins [24,30]. In our study, 86% of the Amb a 11-positive sera displayed IgE reactivity in ELISA toward both the recombinant allergens.

The higher yield of intracellular protein, when compared to secreted (supernatant) protein, was also noted by other groups using the *Sf9* cell line [31,32].

The risk of crossreactivity between Amb a 11 and group 1 mite allergens is low [33]. Indeed, in our study, the ratio of HDM-positive SPT results was actually higher in Amb a 11-negative patients (Table 2), rendering crossreactivity unlikely.

The biological effect of recombinant allergens explored by hRBL mediator release assay performed on selected sera was generally weaker when compared with the nAmb a 1 effect over a wide range of allergen concentrations, but the results were positive and, in some cases, nearly of the same magnitude (sera A and C). This relates to similar in vivo basophil and mast cell degranulation triggered by Amb a 11 interaction with cell-bound IgE antibodies, indicating an important biological activity of this allergen at least in some of the patients. Indeed, QoL is influenced by allergy; sleep disturbances, allergy-related hospitalizations, and high activity impairment scores are present predominantly in Amb a 11-positive patients, without reaching statistical significance when compared to the Amb a 11-negative group (Table 2).

Streptavidin ImmunoCAP appears impaired by the lower sensitivity of this method toward the biotin-conjugated recombinant allergen, with three positive results from the total 44 sera tested, which is at great odds with the ELISA results and could be a consequence of allergen biotinylation. In another study, using biotinylated mouse anti-human IgE for ImmunoCAP and ELISA and comparing the results, it was determined that the latter is more specific (thus, more inclusive for true positive results) but less sensitive [34]. 

The differences registered between the Amb a 11-positive and -negative subgroups warranted further investigation; logistic regression analysis was employed to exclude multiple ragweed pollen components and other environmental exposures as possible confounders. Important confounders, given the sensitization patterns, would be HDM, grass pollen, mold, and cat allergens [35]. Sensitization to Der p 1 (the cysteine protease from HDM) in pre-school children appears to be a marker for developing asthma at school age [36]. Regarding Amb a 11, a murine sensitization model was previously developed, featuring airway inflammation, the production of serum IgEs, and the induction of Th2 immune responses [15]. Other potential confounders would be environmental pollutants (particulates or second-hand smoke) and individual variance in innate mucosal repair mechanisms [37,38,39].

A few variables are retained in the logistic regression model to discriminate between less severe and more severe asthma or asthma-like symptoms (Table 3): IgE reactivity against Amb a 11, positive dog dander SPT, and domestic exposure to cats (in the past 12 months). Amb a 11 IgE sensitization is strongly associated with high symptom scores in these 150 patients (OR = 4.82, 95% CI = 1.81–12.81). The distinction regarding the two pet allergen sources could imply that domestic dog exposure alone is not associated with severe asthma symptoms unless IgE sensitization is manifest (documented by positive dog SPT, OR = 7.48, 95% CI = 2.29–24.46), overcoming immune tolerance. However, cat ownership in the prior 12 months was already proven to be positively associated with allergy manifestations such as wheezing or rhinitis symptoms in children [40]. All the other variables (including IgE sensitization to specific ragweed allergens) added to the model were discarded by the regression algorithm, as they reached *p* > 0.25 (Table 3).

The current study is the first to investigate recombinant mature Amb a 11 production, characterization, and use in mature form, for the assessment of IgE sensitization prevalence and the evaluation of this allergen’s biological activity and clinical significance in a region’s population. Further developments should address small peptide synthesis and IgE-specific epitope mapping using ragweed allergy patients’ sera as preliminary steps in designing personalized immunotherapy.

IgE measurements by ELISA can be influenced by the presence of IgG against the coated protein or by anti-IgE IgG antibodies in patients’ sera [41]. This could be mitigated by specific techniques, but in our case, this would have eventually led only to a small increase in detected IgE prevalence because the patients were displaying allergy symptoms; thus, the immune tolerance had been overwhelmed. Furthermore, natural IgG production appears to be more closely related to chronic spontaneous urticaria (which is not primarily an allergic disease) and not to aeroallergen-induced sensitization (prototype of IgE-mediated type I hypersensitivity reaction) [42,43]. Allergic patients undergoing allergen immunotherapy (AIT) are likely to develop allergen-specific IgG (IgG1 and IgG4 subclasses), which is a desirable marker of immune tolerance, but the selected patients did not follow AIT [44].

The TMA is a part of the greater Banat transnational area, which is inhabited by people with Romanian, Serbian/Croatian, Hungarian, German/Schwaben, Czech/Slovakian, and Bulgarian ancestry. There are distinct differences in HLA typing between this population and other Romanian regions [45]. Thus, IgE sensitization patterns could differ between regions, which means the results might not fully apply to the general Romanian population or other countries, as was already proven in the case of cysteine protease from kiwifruit [13,46,47].

## 5. Conclusions

The present research confirms Amb a 11 as an important IgE-reactive component from ragweed pollen, with significant coverage regarding the specific IgE sensitization (68.67%) and allergenicity of the recombinant allergen comparable to Amb a 1. Sensitization to this major allergen appears to be associated with more severe asthma symptoms (OR = 4.71, 95% CI = 1.81–12.21). Therefore, recombinant Amb a 11 should be included in the panel of relevant allergens from ragweed pollen for effective diagnosis and AIT in ragweed-pollen-allergic patients. 

## Figures and Tables

**Figure 1 biomolecules-13-00182-f001:**
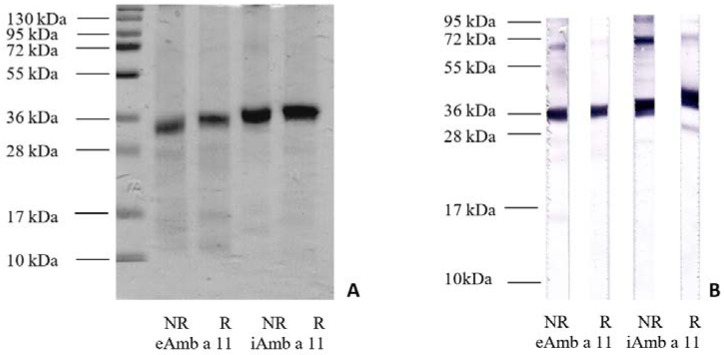
Recombinant allergens overview. (**A**) SDS-PAGE, Coomassie Brilliant Blue R-250-stained and (**B**) His-tag detection, under non-reducing (NR) and reducing (R) conditions.

**Figure 2 biomolecules-13-00182-f002:**
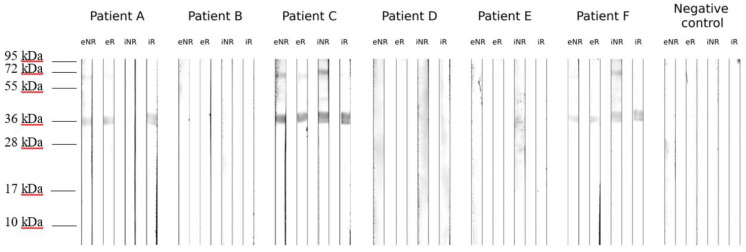
Western blot of eAmb a 11 (eNR and eR) and iAmb a 11 (iNR and iR) against patient sera, in nonreducing (NR) and reducing (R) conditions.

**Figure 3 biomolecules-13-00182-f003:**
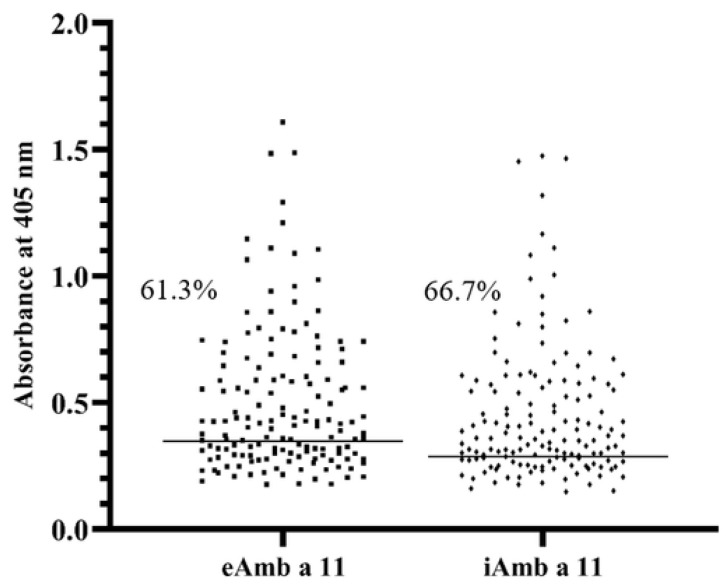
ELISA results for recombinant Amb a 11 IgE binding frequency in ragweed-pollen-allergic patients (the horizontal line indicates the threshold between positive and negative tests).

**Figure 4 biomolecules-13-00182-f004:**
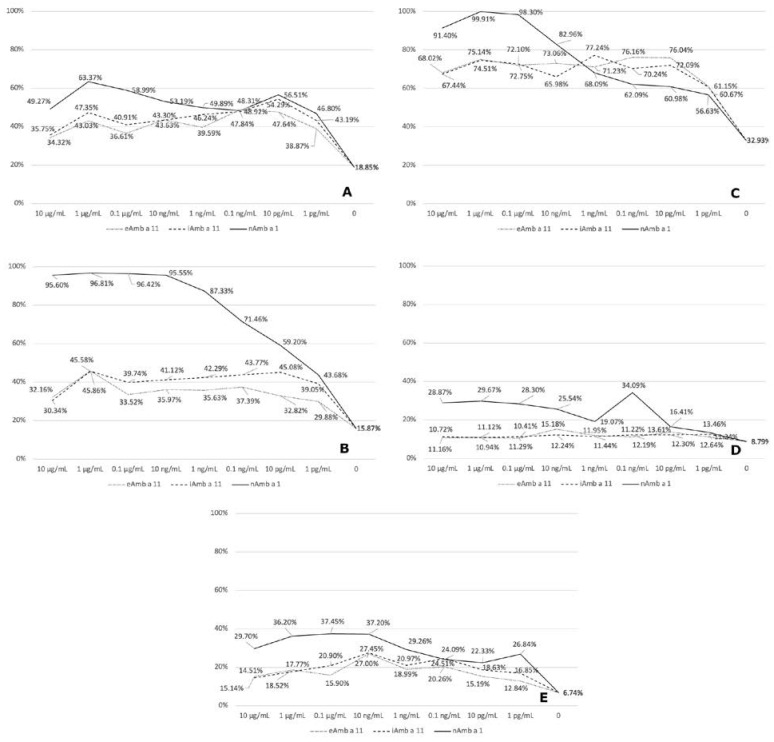
RBL assay in 5 patients (**A**–**E**) with sera that tested positive on ELISA for IgE reactivity toward both recombinant allergen forms. RBL degranulation % is represented on y-axis, upon RBL treatment with a series of allergen concentrations (x-axis). The percentage is relative, compared with total degranulation % (100%) obtained by treating control RBL cells with Triton-X 10%.

**Table 1 biomolecules-13-00182-t001:** Streptavidin ImmunoCAP (pilot run): 44 patient sera tested against biotinylated iAmb a 11.

Patient ID	CAP Test	Concentration (kUA/L) ^†^	Patient ID	CAP Test	Concentration (kUA/L) ^†^
iAmb11-1	o212	0.02	iAmb11-23	o212	0
iAmb11-2	o212	0.03	iAmb11-24	o212	0.01
iAmb11-3	o212	0.01	iAmb11-25	o212	1.83
iAmb11-4	o212	0.01	iAmb11-26	o212	0.46
iAmb11-5	o212	0.01	iAmb11-27	o212	0.05
iAmb11-6	o212	0	iAmb11-28	o212	1.33
iAmb11-7	o212	0	iAmb11-29	o212	0.02
iAmb11-8	o212	0	iAmb11-30	o212	0
iAmb11-9	o212	0.04	iAmb11-31	o212	0
iAmb11-10	o212	0	iAmb11-32	o212	0.01
iAmb11-11	o212	0.01	iAmb11-33	o212	0.01
iAmb11-12	o212	0.1	iAmb11-34	o212	0.12
iAmb11-13	o212	0	iAmb11-35	o212	0
iAmb11-14	o212	0.04	iAmb11-36	o212	0.01
iAmb11-15	o212	0.03	iAmb11-37	o212	0.03
iAmb11-16	o212	0	iAmb11-38	o212	0.02
iAmb11-17	o212	0.02	iAmb11-39	o212	0.03
iAmb11-18	o212	0.01	iAmb11-40	o212	0.03
iAmb11-19	o212	0.01	iAmb11-41	o212	0.01
iAmb11-20	o212	0.01	iAmb11-42	o212	0.01
iAmb11-21	o212	0.02	iAmb11-43	o212	0.01
iAmb11-22	o212	0	iAmb11-44	o212	0.01

^†^ Positive ImmunoCAP result if concentration > 0.35 kUA/L.

**Table 2 biomolecules-13-00182-t002:** Descriptive statistics of the patients’ dataset, with statistical comparison between Amb a 11-positive and -negative subsets.

Parameters ^†^	Patients Group (*n* = 150)	Amb a 11 Positive (*n* = 103)	Amb a 11 Negative (*n* = 47)	*p* Value
Age (years)	35.91 ± 8.75 (18–61, 35)	36.04 ± 8.93 (18–61, 35)	35.62 ± 8.43 (20–61, 34)	>0.05 NS
Age groups	18–39 years: 106 (70.67%) 40–54 years: 39 (26%) ≥55 years: 5 (3.33%)	18–39 years: 73 (70.88%) 40–54 years: 27 (26.21%) ≥55 years: 3 (2.91%)	18–39 years: 33 (70.21%) 40–54 years: 12 (25.53%) ≥55 years: 2 (4.26%)	>0.05 NS
Sex	Women: 50 (33.33%) Men: 100 (66.67%)	Women: 38 (36.89%) Men: 65 (63.11%)	Women: 12 (25.53%) Men: 35 (74.47%)	**<0.05 S**
Allergy history (years)	5.06 ± 4.96 (1–32, 3)	5.28 ± 4.84 (1–24, 3)	4.56 ± 5.23 (1–32, 3)	>0.05 NS
Family history of allergic disease	Negative: 107 (71.33%) Positive: 43 (28.67%)	Negative: 72 (69.9%) Positive: 31 (30.1%)	Negative: 35 (74.47%) Positive: 12 (25.53%)	>0.05 NS
**Exposure factors**				
Smoking	Non-smokers: 89 (59.33%) Smokers: 61 (40.67%)	Non-smokers: 61 (59.22%) Smokers: 42 (40.78%)	Non-smokers: 28 (59.57%) Smokers:19 (40.43%)	>0.05 NS
Professional (organic solvents, dust, etc.)	Negative: 122 (81.33%) Positive: 28 (18.67%)	Negative: 84 (81.55%) Positive: 19 (18.45%)	Negative: 38 (80.85%) Positive: 9 (19.15%)	>0.05 NS
Pets	Negative: 96 (64%) Positive: 54 (36%)	Negative: 71 (68.93%) Positive: 32 (31.07%)	Negative: 25 (53.19%) Positive: 22 (46.81%)	**<0.05 S**
**Clinical data**				
Allergy pattern	Monosensitized: 45 (30%) Polysensitized: 105 (70%)	Monosensitized: 38 (36.89%) Polysensitized: 65 (63.11%)	Monosensitized: 7 (14.89%) Polysensitized: 40 (85.11%)	**<0.05 S**
Additional positive skin prick test results	House dust mite: 54 (36%) Cereal/grass pollen: 48 (32%) Artemisia: 42 (28%) Fungi: 27 (18%) Tree pollen: 25 (16.67%)	House dust mite: 35 (33.98%) Cereal/grass pollen: 26 (25.24%) Artemisia: 24 (23.3%) Fungi: 15 (14.56%) Tree pollen: 13 (12.62%)	House dust mite: 19 (40.43%) Cereal/grass pollen: 22 (46.81%) Artemisia: 18 (38.30%) Fungi: 12 (25.53%) Tree pollen: 12 (25.53%)	>0.05 NS **<0.05 S** **<0.05 S** **<0.05 S** **<0.05 S**
Allergy season (months)	3.24 ± 1.86 (1–12, 3)	3.06 ± 1.74 (1–12, 2)	3.66 ± 2.06 (1–12, 3)	>0.05 NS
Symptoms score (on a 1–12 scale for rhinitis/asthma, 1–9 scale for conjunctivitis/urticaria) and frequency of disease	Rhinitis: 7.27 ± 2.80 (1–12, 8) 149 patients (99.33%) Conjunctivitis: 4.58 ± 2.3 (1–9, 4) 136 patients (90.67%) Asthma: 3.47 ± 2.38 (1–11, 3) 90 patients (60%) Urticaria: 3.11 ± 1.87 (1–9, 3) 37 patients (24.67%) Edema: 13 patients (8.67%)	Rhinitis: 7.30 ± 2.88 (1–12, 8) 103 patients (100%) Conjunctivitis: 4.6 ± 2.43 (1–9, 4) 95 patients (92.23%) Asthma: 3.84 ± 2.54 (1–11, 3) 63 patients (61.17%) Urticaria: 3.17 ± 2.04 (1–9, 3) 24 patients (23.3%) Edema: 8 patients (7.77%)	Rhinitis: 7.20 ± 2.66 (1–12, 8) 46 patients (97.87%) Conjunctivitis: 4.54 ± 2.01 (1–9, 5) 41 patients (87.23%) Asthma: 2.59 ± 1.69 (1–6, 2) 27 patients (57.45%) Urticaria: 3 ± 1.58 (1–6, 3) 13 patients (27.66%) Edema: 5 patients (10.64%)	>0.05 NS >0.05 NS >0.05 NS >0.05 NS **<0.05 S** >0.05 NS >0.05 NS >0.05 NS >0.05 NS
**Quality of Life indicators**				
Sleep disturbance	Absent: 48 (32%) Present: 102 (68%)	Absent: 31 (30.1%) Present: 72 (69.9%)	Absent: 17 (36.17%) Present: 30 (63.83%)	>0.05 NS
Allergy-related hospitalization	Absent: 133 (88.67%) Present: 17 (21.33%)	Absent: 88 (85.44%) Present: 15 (14.56%)	Absent: 45 (95.74%) Present: 2 (4.26%)	>0.05 NS
Activity impairment score (1–10 scale)	6.88 ± 2.62 (1–10, 7.5)	7.09 ± 2.60 (1–10, 8)	6.43 ± 2.64 (1–10, 7)	>0.05 NS

^†^ Numerical variables presented as: mean ± standard deviation (minimum–maximum and median). Categorical variables presented as: frequency (cases) and percentage.

**Table 3 biomolecules-13-00182-t003:** Logistic regression results: odds ratios (ORs) and 95% confidence intervals (CIs)—complete analysis available in Appendix A.

Variable	Odds Ratio	95% CI
IgE sensitization to Amb a 10	1.8279	0.6732 to 4.9630
IgE sensitization to Amb a 11	4.8207	1.8128 to 12.8193
IgE sensitization to Amb a 6	0.5092	0.2175 to 1.1922
Positive dog SPT	7.4861	2.2907 to 24.4646
Positive fungi SPT	0.3569	0.1103 to 1.1549
Domestic exposure to cats	3.3427	1.2917 to 8.6502

## Data Availability

Not applicable.

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
