# Peer review of "Ragweed Major Allergen Amb a 11 Recombinant Production and Clinical Implications"

_biomolecules, 2023, doi:10.3390/biom13010182_

Round 1

Reviewer 1 Report

The manuscript “ Ragweed major allergen Amb a 11 recombinant production and clinical implications” by Tamas and collaboratores is focused on the comparative immunological analysis of recombinant ragweed allergen Amb a 11 expressed in E. coli and Spodoptera frugiperda. Furthermore, the biological activity of the recombinant allergens has been investigated.

Comments

Although we can appreciate the authors efforts for the work done, is not clearly written and many inaccuracies are present throughout the text. The article in this form is not suitable for publication.

Just some example 

In the Introduction section the Authors do not make a presentation that corresponds sufficiently to the title of the article, the topic is not well framed. Therefore I suggest to the Authors to focus a little more on the description of the Amb a 11 as belonging to the family of cysteine proteases.

The Results section does not provide sufficient details of the results obtained by experimental procedures adopted, moreover, due to their small size, some figures presented do not allow an accurate assessment by the reviewer. 

The Results are reported in the Discussion section, for example lines:

-322-325 “SDS-PAGE, together with gel filtration evidence of 2 protein fractions (Figure, un-322 published material), suggest that Amb a 11 in both expression forms can aggregate pref-323 erentially as dimers, which display the hexahistidine tag on the surface and retain IgE 324 reactivity when tested against patient’s sera.”

-338-339 “In our case, the  Amb a 11 protein appears to be slightly below iAmb a 11 on gel, at 36 kDa.” I suggest to the Authors to discuss in detail the results in the SDS-PAGE the low molecular weight bands that are visible in both reducing and non-reducing conditions and so on.

Author Response

Dear reviewer,

Thank you very much for taking the time to check this work. In the introductory part I added some information on current knowledge on the protein Amb a 11. Regarding the overall article structure, I consolidated it by grouping related methods (Western blot based: His tag detection together with IgE reactivity assessment for the recombinant allergens). Also consolidated all the statistics in the later part of the results and discussions. 

In the results indeed there was little description - this has been corrected, especially for the gel and western blot. In the discussion section maybe you will find the proposed interpretation interesting. 

I also rearranged the figure 4, hope it is more readable now.

Best regards,

Paul Tamas, MD

Reviewer 2 Report

In this manuscript, the authors express one of Ragweed's pollen allergens in E. coli and insect cells and extensively study its reactivity with patient sera. Since information on this allergen is scarce, this manuscript should provide important insights. We hope that the manuscript can be improved in several areas as follows.

1. A list of Ragweed allergens (including name, molecular weight, properties, attribution, etc.) would be helpful to the reader.

2. The term “immunoblot” and the term “western blot” are mixed in the text. Do you use these different meanings? If so, you need to explain how you are using them.

3. Some words that should be subscripted are not subscripted. ( L.169, L.346)

4. Fig.4 is too small and thin and needs to be improved.

5. Is “Ryousuke Nakamura” a Prof. in L.178?  I think he is a researcher of the institute, not a Prof., so Dr. is appropriate. Please confirm.

6. What are L193 and “Gold buffer”?  Explanation is needed.

Author Response

Dear reviewer,

Thank you very much for taking the time to check this work. Regarding the overall article structure, I consolidated it by grouping related methods (Western blot based: His tag detection together with IgE reactivity assessment against patient sera, for the recombinant allergens). Also consolidated all the statistics in the later part of the results and discussions. 

On the points you mentioned

1) A list of ragweed allergens would be useful indeed, however I am not sure how to integrate it as they are many ragweed allergens. The list would be long, not very suitable for introduction part, and it would make the article look maybe like a review. For the moment I added some descriptive elements on the protein allergen in the introduction.

2) Immunoblot and western blot are also found in literature with the same or very similar meaning, but I am sticking with western blot now and removed the immunoblot for ease of reference.

https://www.sigmaaldrich.com/RO/en/technical-documents/protocol/protein-biology/western-blotting/western-blotting

3) The missing subscripts are double and triple checked now

4) I rearranged the elements in figure 4, to be oriented vertically and fit better in page. Hopefully it is more readable now

5) That is a mistake indeed, he is not appearing as a professor according to a Japanese professionals web site. In my defense, I also found a Nature article citing him as a professor.

https://jglobal.jst.go.jp/en/detail?JGLOBAL_ID=200901047924690644

6) I made sure Gold buffer composition is explained in the first time it appears in text. 

Thank you again, and hope to hear from you soon !

Paul Tamas, MD

Round 2

Reviewer 1 Report

I really appreciated the efforts of the Authors in addressing the criticisms raised by the rewiever. The manuscript can be considered for publication in Biomolecules.

Best regards